# Consistent Correlation between MTHFR and Vascular Thrombosis in Neonates—Case Series and Clinical Considerations

**DOI:** 10.3390/jcm12144856

**Published:** 2023-07-24

**Authors:** Catalin Cirstoveanu, Nicoleta Calin, Carmen Heriseanu, Cristina Filip, Corina Maria Vasile, Irina Margarint, Veronica Marcu, Mihai Dimitriu, Liliana Ples, Sorin Tarnoveanu, Mihaela Bizubac

**Affiliations:** 1Department of Neonatal Intensive Care, “Carol Davila” University of Medicine and Pharmacy, 020021 Bucharest, Romania; catalin.cirstoveanu@umfcd.ro (C.C.); mihaela.bizubac@drd.umfcd.ro (M.B.); 2Neonatal Intensive Care Unit, “M.S. Curie” Children’s Hospital, Constantin Brâncoveanu Boulevard, No. 20, 4th District, 041451 Bucharest, Romania; 3Ph.D. School Department, “Carol Davila” University of Medicine and Pharmacy, 020021 Bucharest, Romania; irina-maria.margarint@drd.umfcd.ro (I.M.); mihai.dimitriu@umfcd.ro (M.D.); liana.ples@umfcd.ro (L.P.); 4Department of Pediatrics, “Carol Davila” University of Medicine and Pharmacy, 020021 Bucharest, Romania; cristina.filip@umfcd.ro; 5Pediatric Cardiology, “M.S. Curie” Children’s Hospital, Constantin Brâncoveanu Boulevard, No. 20, 4th District, 041451 Bucharest, Romania; corina.vasile93@gmail.com; 6Department of Pediatric and Adult Congenital Cardiology, University Hospital of Bordeaux, 33600 Pessac, France; 7Pediatric Cardiovascular Surgery, “M.S. Curie” Children’s Hospital, Constantin Brâncoveanu Boulevard, No. 20, 4th District, 041451 Bucharest, Romania; 8Department of Radiology, “M.S. Curie” Children’s Hospital, Constantin Brâncoveanu Boulevard, No. 20, 4th District, 041451 Bucharest, Romania; radiologie@ms.ro; 9“Sf. Pantelimon” Emergency Clinical Hospital, 340-342 Pantelimon Road, 021661 Bucharest, Romania; 10“Bucur” Maternity, “Saint John” Hospital, Intre Garle Street, 040294 Bucharest, Romania; 11Department of Neurosurgery, “M.S. Curie” Children’s Hospital, Constantin Brâncoveanu Boulevard, No. 20, 4th District, 041451 Bucharest, Romania; tarnoveanu@yahoo.com

**Keywords:** MTHFR mutations, venous thrombosis, autosomal recessive pattern, case series

## Abstract

Background: MTHFR polymorphism has been inconsistently linked to thrombotic events—some studies have shown its contribution to venous thrombosis, arterial thrombosis, and ischemic stroke, whereas others have found no statistically significant correlation between them. Methods: A descriptive case series study was performed in the Neonatal Intensive Care Unit of “Marie Sklodowska Curie” Emergency Clinical Hospital for Children in Bucharest, Romania. Results: All patients had positive results for MTHFR variants; 14 patients were positive for compound heterozygosity, 13 patients for MTHFR C677T (seven of which were homozygous), and 13 patients for MTHFR A1298C (three of which were homozygous). Eighteen patients received anticoagulants (heparin, enoxaparin, or bivalirudin), and thrombolytics (alteplase) were administered in six cases. In one case, a thrombectomy was performed; in another, vascular plasty was undertaken. Only in six cases was complete revascularization possible. Incomplete revascularization occurred for one patient with a negative outcome. Conclusion: The particularity of this case series is that every patient in our unit who developed thrombi had a positive genetic result for MTHFR mutations. MTHFR mutations should be regarded as a thrombotic risk factor for critically ill patients, and screening for MTHFR mutations should be performed in every admitted patient to intensive care units, thus achieving the prevention of thrombi.

## 1. Introduction

Methylenetetrahydrofolate reductase (MTHFR) is an enzyme in the folate cycle that reduces 5,10-methylenetetrahydrofolate—the major form of cellular folate—to 5-methyltetrahydrofolate, which is the main form of circulating folate [1] and the substrate for methionine synthase, serving as a methyl donor in the methionine cycle. This leads to the methyl-cobalamin (methyl-vitamin B12)-dependent conversion of homocysteine, a sulfur amino acid, to methionine, an essential amino acid. Subsequently, methionine provides the methyl group for further reactions, being transformed into S-adenosylmethionine, which plays an important role in the trans-sulphuration pathway and various trans-methylation reactions [2] (Figure 1). Therefore, MTHFR catalyzes vital cell metabolism processes, including DNA and RNA synthesis and repair and protein methylation, providing a normal range of methionine and homocysteine [3,4].

The MTHFR gene is found on the short arm of chromosome 1 (1p36.3). Being inherited in an autosomal recessive pattern, mutations of this gene are known as the most frequent errors in folate metabolism [4]. C677T and A1298C are those most commonly described as point mutations causing amino acid substitutions [5].

MTHFR C677T polymorphism occurs on exon 4 in the N-terminal catalytic domain, exchanging the 677 base-pair cytosine for thymine, thus replacing alanine with valine on codon 222 [6,7]. This causes a thermolability of the MTHFR enzyme in homozygotes (TT genotype); its activity is reduced to 40–50% at 37 °C, decreasing to 35% at higher temperatures of 46 °C [5], leading to lower plasma levels of 5-methyltetrahydrofolate, increased plasma levels of 5,10-methylenetetrahydrofolate, and moderately increased levels of homocysteine, especially in the presence of low folate levels in plasma [2].

In heterozygotes (CT genotype), reports showed a mean MTHFR enzyme activity of 65%. However, at higher temperatures of 46 °C, its activity only decreases to 56% [6] and only a mild increase in homocysteine has been reported [2,3,5].

MTHFR A1298C polymorphism occurs on exon 7 in the C-terminal regulatory domain [7], exchanging the 1298 base-pair adenosine to cytosine, thus replacing glutamic acid to alanine on codon 429. This causes a reduced activity of MTHFR function, down to 68% for homozygotes, although maintaining its thermostability and normal levels of folate and homocysteine in both homozygotes and heterozygotes [1,3].

Double heterozygotes for both C677T and A1298C polymorphisms are reported to have a 50–60% enzyme activity [7], low plasma folate concentrations, and high homocysteine levels [1,3]. Homocysteine levels have been reported to be more elevated than in the homozygous C677T mutation. No thermolability has been observed in combined heterozygosity [1].

Other rare mutations of the MTHFR gene have been described to result in a critical reduction of the enzyme activity, between 0 and 20%, causing highly elevated levels of homocysteine (more than 10 times the normal value), which subsequently leads to homocystinuria [2,5]. Low levels of methionine and circulating folates have also been reported [3,4].

The prevalence of TT homozygous C677T mutations varies geographically, with the highest values of 32% in the Mexican population and the lowest (less than 2%) in Afro-American and sub-Saharan African populations [3,5,8]. The prevalence of homozygous A1298C mutations is 9–10% in Caucasian people. Double heterozygosity for C677T and A1298C polymorphisms has a 15–20% prevalence [3,7,9].

In adults, MTHFR C677T polymorphism is especially associated with vascular diseases, such as hemorrhagic or ischemic stroke, coronary artery disease, essential hypertension, and hyperlipidemia. A weak association has also been described between MTHFR C677T mutation, carotid dissection, retinal vein occlusion, and venous thromboembolism. Moreover, neoplastic diseases have been directly associated with MTHFR C677T polymorphism due to folate deficiency and insufficient DNA methylation, causing tumorigenesis. A higher risk of relapse for acute lymphocytic leukemia and an increased risk for diabetic nephropathy have also been described [3,5]. Other reports have shown a significant association between MTHFR C677T variant and infertility, recurrent pregnancy loss, and other obstetrical complications, including severe pre-eclampsia, abruptio placentae, fetal growth retardation, and stillbirth [2,5,10]. A minimal association was also found between MTHFR C677T mutation and psoriasis vulgaris due to methylation deficiency. Furthermore, due to folate and vitamin B12 deficiencies, neurologic and psychiatric diseases are indirectly associated with MTHFR C677T polymorphism, including Parkinson’s disease, Alzheimer’s disease, migraines, and combined psychiatric disorders [5].

In children, an increased risk for ischemic stroke has been reported in association with MTHFR C677T mutation [5]. Limited findings showed that Down syndrome could also be linked to the MTHFR C677T variant [2,11], especially in maternal polymorphism, due to abnormal folate metabolism [5]. Still, more data are needed on this matter [9]. Additionally, MTHFR C677T mutation has been discovered to have a protective effect in children with acute lymphocytic leukemia, as opposed to adults [5]. Rare MTHFR mutations that cause severe enzyme activity deficiency are usually associated with neuropathy, encephalopathy, vasculopathy, and coagulopathy. Symptoms typically present in infants or teenagers include development delay, several neurologic and psychiatric conditions, periodic thrombosis, and thromboembolism [3,6].

As for neonates, neural tube defects have been frequently associated with MTHFR C677T mutation [2,5,11,12] and its prevalence following the homozygous TT genotype geographical distribution [8]. C677T homozygosity and compound heterozygosity in the mother increase the risk of developing neural tube defects [9]. MTHFR A1298C may also be associated with neural tube defects, especially in the presence of low folate levels [1], but this mutation alone has no significant increase in the risk [9]; recent meta-analysis suggested no association between these conditions [13]. Orofacial clefts, congenital heart defects, and fetal anticonvulsant syndrome could also be associated with MTHFR C677T mutation, but more data are required [2,9,11].

While other inherited thrombophilia mutations, such as factor V Leiden and prothrombin G20210A, as well as hyperhomocysteinemia, are generally recognized for their important contribution to developing a hypercoagulable state—which could lead to venous thromboembolism (deep vein thrombosis, pulmonary embolism, cerebral vein thrombosis, thrombosis in the portal, mesenteric or hepatic veins), thrombosis of placental vessels, stroke or transient ischemic attack, myocardial infarction, fetal loss and obstetric complications [10,14,15]—MTHFR polymorphism has been inconsistently linked to thrombotic events. Some studies have shown its contribution to venous thrombosis [16,17,18,19,20], arterial thrombosis [21,22], and ischemic stroke [23,24]. In contrast, others have found no statistically significant correlation [6,25,26,27,28,29,30,31].

## 2. Materials and Methods

### 2.1. Study Design, Setting, and Population

We conducted a retrospective, descriptive case series study, which included 40 patients admitted to the Neonatal Intensive Care Unit from “Marie Sklodowska Curie” Emergency Clinical Hospital for Children in Bucharest, Romania, over eleven years, between October 2011 and February 2023. All patients presented thrombotic events in various body regions and had positive genetic results for MTHFR variants.

### 2.2. Data Collection

We collected data from the available medical records, including paraclinical investigations, medical imaging, and patient history, since birth and until discharge from our unit. The available information was analyzed and compiled into a case series.

### 2.3. Statistical Analysis

The obtained data were processed and interpreted using Microsoft Excel 2007.

### 2.4. Aim

Our objective was to analyze and present the thrombotic events in patients who tested positive for the MTHFR genetic variant and the diverse imaging findings associated with these events. Additionally, we aimed to explore the role of other known pro-thrombotic conditions and their contribution to the overall understanding of thrombosis in these patients.

## 3. Results

In our unit, a total of 40 patients were admitted between the first day of life and 10 weeks of life. Among them, 24 were males and 16 were females. Out of the 40 cases, 18 were admitted due to thrombotic-related conditions. Additionally, 11 patients were diagnosed with in utero thrombosis before birth. In five cases, the exact timing of the thrombosis remained uncertain due to insufficient data before the transfer. The onset of symptoms occurred at home for four patients, and the thrombotic event recurred in eight cases. Among the mothers, eight tested positive for congenital thrombophilia, with four being tested after the patient was diagnosed. Furthermore, six mothers reported experiencing at least one miscarriage before giving birth, and three had a SARS-CoV-2 infection during pregnancy. Four patients had a positive family history of thrombosis or congenital thrombophilia, excluding the mother’s history (as shown in Table 1).

In our study, thrombosis affected multiple anatomical regions of the body, as illustrated in Figure 2. Among the cases analyzed, 20 patients presented with intracranial thrombosis, involving conditions such as intraventricular or intraparenchymal hemorrhages, cerebellar hemorrhages, post-hemorrhagic hydrocephalus, cerebral venous sinuses thrombosis, and ischemic strokes. Additionally, ten patients exhibited thrombosis of the abdominal vessels, including the inferior vena cava, portal vein, renal veins, common iliac veins, external iliac veins, mesenteric veins, abdominal aorta, common iliac arteries, and mesenteric arteries.

Moreover, five patients displayed thrombosis in the limbs or peripheral regions, leading to ischemia and necrosis of the skin, as well as involvement of the femoral and axillary arteries. Four patients demonstrated thoracic thrombosis, encompassing intracardiac thrombosis, pulmonary arteries, and chylothorax. Finally, two patients presented with generalized thrombosis, affecting the internal jugular veins, brachiocephalic veins, both venae cavae, common iliac veins, external iliac veins, and femoral veins.

All patients had positive results for MTHFR variants, 14 patients were positive for compound heterozygosity, 13 patients for MTHFR C677T (seven of which were homozygous), and 13 patients for MTHFR A1298C (three of which were homozygous). Positive genetic results for other congenital thrombophilia mutations were identified, as well: Factor V Leiden, Prothrombin G20210A, Factor XIII (Val34Leu), Fibrinogen 455 G>A, GP IIb/IIIa L33P and Factor V A4070G (HR2 haplotype) heterozygosity, PAI-1 4G/5G promoter variants, and activated protein C resistance V. In addition, 28 patients had associated protein S, protein C, or antithrombin deficiency. Normal homocysteine levels were detected in ten cases—three MTHFR C677T homozygosity cases and five double heterozygosity cases. After receiving the positive genetic result, the family underwent screening for congenital thrombophilia in only five cases, all with positive results (Table 2).

Eighteen patients received anticoagulants (heparin, enoxaparin, or bivalirudin), and thrombolytics (alteplase) were administered in six cases [32,33,34]. Two patients developed thrombosis during anticoagulation therapy for other conditions—hemodialysis and Blalock–Taussig shunt, whereas two other patients developed generalized thrombosis while being anticoagulated for the initial event. In one case, a thrombectomy was performed; in another, vascular plasty was undertaken. Only in six cases was complete revascularization possible. Incomplete revascularization occurred for one patient with a negative outcome. Three patients also received vasoactive agents like pentoxifylline or alprostadil. Only two patients were administered vitamin supplementation with thiamin, pyridoxine, or folic acid. Finally, 11 patients had a negative outcome, 28 were discharged, and 1 is still hospitalized.

## 4. Serial Case Reports

### 4.1. Case No. 1—Home Onset—Left Intraventricular and Intraparenchymal Hemorrhage

A 2-week-old female neonate (39 weeks, 3.16 kg) presented with opisthotonos and generalized tonic–clonic seizures, known with hydrocephalus of the lateral and third ventricles, from a consultation performed a couple of hours before admission. The brain ultrasound from admission indicated left intraventricular hemorrhage and intraparenchymal hemorrhage of the left parietal and occipital lobes, affecting the left thalamic and subthalamic regions, with extension to the right thalamus (Figure 3).

A suspicion of postnatal stroke was raised, and genetic testing confirmed inherited thrombophilia, with positive MTHFR A1298C and Fibrinogen 455 G>A heterozygous gene mutations and Protein C deficiency. There was limited information about the family history.

### 4.2. Case No. 2—In Utero Right Intraventricular Hemorrhage, Post-Hemorrhagic Hydrocephalus

A 7-day-old female neonate (38 weeks, 3.11 kg), known since the 36th week of gestation with important rapidly progressive post-hemorrhagic hydrocephalus, presented with bulging anterior fontanelle and macrocephaly.

Genetic testing confirmed inherited thrombophilia, with positive Factor V Leiden, Factor V A4070G (HR2 haplotype), MTHFR A1298C, PAI-1 4G/5G promoter, Factor XIII (Val34Leu) and Fibrinogen 455 G>A heterozygous gene mutations. There was no known family history of thrombophilia or miscarriage.

### 4.3. Case No. 3—Perinatal Stroke

A 10-week-old female infant (36 weeks, 2.70 kg) known from birth with generalized tonic–clonic seizures, absence of spontaneous breathing, fixed bilateral mydriasis, and coma, presented for hypoxic–ischemic encephalopathy. The admission laboratory findings showed an elevated c-reactive protein. A brain CT scan was performed, identifying the hyperdensity of the thalamic nuclei. Treatment with alteplase was initiated.

A suspicion of perinatal stroke was raised, and genetic testing confirmed inherited thrombophilia, with positive MTHFR A1298C and MTHFR C677T heterozygous gene mutations. There was no known family history of thrombophilia. Following this diagnosis, maternal screening was performed, yielding similar results. The mother confirmed experiencing reduced fetal movement one week before giving birth.

### 4.4. Case No. 4—Repeated Postnatal Intraparenchymal and Intraventricular Hemorrhage

A 2-week-old male neonate (36 weeks, 2.99 kg), delivered by emergency cesarean section due to abnormal cardiotocography and severe oligohydramnios, with true umbilical cord knot at birth, presented with left intraventricular and intraparenchymal hemorrhage revealed from the brain ultrasound performed at 8 h of life. Myoclonic seizures appeared three days after admission, progressing to generalized tonic–clonic seizures. An ultrasound of the brain was performed, which indicated new bleeding of the bilateral basal ganglia (Figure 4).

Genetic testing confirmed inherited thrombophilia, with positive MTHFR A1298C and MTHFR C677T heterozygous gene mutations and Antithrombin, Protein S, and Protein C deficiencies. Normal homocysteine levels were detected. There was a positive family history of maternal thrombophilia, treated with enoxaparin and acetylsalicylic acid during pregnancy.

### 4.5. Case No. 5—In Utero Right Intraventricular Hemorrhage, Post-Hemorrhagic Hydrocephalus

A 1-day-old female neonate (37 weeks, 3.44 kg) presented with post-hemorrhagic hydrocephalus due to right fetal intraventricular hemorrhage (fetal MRI performed at 34 weeks of gestation). On admission, an elevated c-reactive protein was detected. An ultrasound of the brain showed hydrocephalus of the lateral, third, and fourth ventricles and a 13/10 mm thrombus inside the right lateral ventricle.

Genetic testing confirmed inherited thrombophilia, with a positive MTHFR A1298C homozygous gene mutation and Protein C deficiency. Normal homocysteine levels were detected. There was no known family history of thrombophilia. Family screening for thrombophilia was performed afterwards, with a positive result for the mother, the father, and the grandmother (with a history of recurrent migraines).

### 4.6. Case No. 6—In Utero Post-Hemorrhagic Hydrocephalus

A 2-day-old male neonate (35 weeks, 3.05 kg) presented with post-hemorrhagic hydrocephalus, known since the 30th week of gestation.

A suspicion of fetal stroke was raised and genetic testing confirmed inherited thrombophilia, with positive MTHFR A1298C and MTHFR C677T heterozygous gene mutations and Antithrombin, Protein S, and Protein C deficiencies. Normal homocysteine levels were detected. There was limited information about the family history.

### 4.7. Case No. 7—In Utero Cerebellar Hemorrhage, Post-Hemorrhagic Hydrocephalus, and Postnatal Intraparenchymal Hemorrhage

A 1-day-old female neonate (35 weeks, 2.25 kg) presented with post-hemorrhagic hydrocephalus affecting the lateral and third ventricles. This condition had been known since the 33rd week of gestation, as the 28-week gestational age scan had shown no abnormalities. A fetal MRI indicated stenosis of the Sylvius aqueduct caused by cerebellar hemorrhage, especially of the vermis, and global cerebellar atrophy. The admission laboratory findings showed modified coagulation tests. During hospitalization, a brain ultrasound showed new intraparenchymal hemorrhages of both frontal and left parietal lobes.

Genetic testing confirmed inherited thrombophilia, with a positive Factor V Leiden heterozygous gene mutation, MTHFR C677T homozygous gene mutation, and Protein S and Protein C deficiencies. Normal homocysteine levels were detected. There was no known family history of thrombophilia (healthy second twin), but a positive history of gestational hypertension (GA of 29–30 weeks). Maternal screening for thrombophilia was performed afterward and MTHFR A1298C and MTHFR C677T gene mutations were detected.

### 4.8. Case No. 8—Perinatal Intraparenchymal Hemorrhage

A 5-week-old female infant (35 weeks, 2.10 kg) known from birth with jaundice of unknown etiology and seizures presented with intense yellowing of the skin and eyes and hypotonia. A brain CT scan performed at 8 days of life identified intraparenchymal hemorrhage of the right parietal and occipital lobes. On admission, a brain ultrasound revealed an old intraparenchymal hemorrhage and later porencephaly in the right parietal and occipital lobes.

Genetic testing confirmed inherited thrombophilia, with positive PAI-1 4G/5G promoter and MTHFR C677T homozygous gene mutations and Antithrombin, Protein S, and Protein C deficiencies. Normal homocysteine and methionine levels were detected. There was a positive family history of maternal thrombophilia (PAI-1 4G/5G promoter, MTHFR A1298C, and MTHFR C677T heterozygous gene mutations). Intrauterine growth restriction and gestational hypertension were also reported.

### 4.9. Case No. 9—Perinatal Intraventricular Hemorrhage and Hydrocephalus

A 3-week-old male neonate (36 weeks, 2.92 kg), known from birth with hypotonia, hydrocephalus of all four ventricles and periventricular leukomalacia on brain ultrasound, presented with bulging anterior fontanelle and macrocephaly. The laboratory findings from birth showed highly elevated presepsin. An MRI of the brain revealed important encephalomalacia, especially in the supratentorial regions of the brain, signs of intraventricular hemorrhage, and nonobstructive hydrocephalus of all four ventricles.

Genetic testing confirmed inherited thrombophilia, with positive MTHFR A1298C and MTHFR C677T heterozygous gene mutations and Antithrombin and Protein C deficiencies. There was no known family history of thrombophilia and no history of miscarriage.

### 4.10. Case No. 10—Postnatal Right Intraventricular and Intraparenchymal Hemorrhage and Sublingual Hematoma

A 1-day-old male neonate (42 weeks, 2.15 kg) presented with VACTERL disorder (vertebral defects and numerical rib anomalies, esophageal atresia with distal tracheoesophageal fistula, and horseshoe kidney). Surgical correction of the esophageal atresia was performed at six days of life. At 18 days of life, the patient developed seizures and opisthotonos. An ultrasound of the brain indicated right intraventricular hemorrhage and intraparenchymal hemorrhage of the right basal ganglia and thalamus. An elevated, but decreasing c-reactive protein was identified. Later, during hospitalization, a sublingual hematoma was noticed (Figure 5).

Suspicion for inherited thrombophilia was raised. Genetic testing confirmed it, with positive Prothrombin G20210A, PAI-1 4G/5G promoter, MTHFR A1298C, and MTHFR C677T heterozygous gene mutations and Antithrombin, Protein S, and Protein C deficiencies. Normal homocysteine and methionine levels were detected. Subsequently, the patient received thiamin and pyridoxine supplementation, tripling the needed folic acid. There was no known family history of thrombophilia or miscarriages.

### 4.11. Case No. 11—In Utero Post-Hemorrhagic Hydrocephalus

A 13-day-old male neonate (36 weeks, 3.05 kg) presented with esophageal atresia, distal tracheoesophageal fistula, and severe congenital hydrocephalus (ventriculo-amniotic shunting performed in utero).

Fetal hemorrhagic stroke was suspected. Genetic testing confirmed inherited thrombophilia, with a positive PAI-1 4G/5G promoter homozygous gene mutation, MTHFR A1298C and MTHFR C677T heterozygous gene mutations, and Protein S and Protein C deficiencies. There was no known family history of thrombophilia.

### 4.12. Case No. 12—Postnatal Thrombosis of the Left Transverse Cerebral Venous Sinus

A 3-hour-old male neonate (38 weeks, 3.69 kg) presented with a left-sided congenital diaphragmatic hernia known since the third trimester of pregnancy to undergo emergency surgical repair. The admission laboratory findings detected leukocytosis, modified coagulation tests, and a mild elevation of the D-dimer tests. Treatment with enoxaparin was initiated, but the D-dimer values continued to increase to undetectable levels, and enoxaparin was switched to continuous heparin infusion. During that period, a routine head ultrasound revealed thrombosis of the left transverse cerebral venous sinus, which disappeared after two weeks as the D-dimer tests decreased.

Genetic testing confirmed inherited thrombophilia, with positive Factor V Leiden and MTHFR A1298C heterozygous gene mutations, Activated Protein C Resistance V, and Antithrombin, Protein S, and Protein C deficiencies. A moderate elevation of PAI-1 was also detected. There was no known family history of thrombophilia or miscarriages.

### 4.13. Case No. 13—Postnatal Intraparenchymal Hemorrhage

A 16-day-old male neonate (30 weeks, 1.62 kg) presented with a patent arterial duct unresponsive to pharmacological closure. During hospitalization, the patient underwent several invasive procedures: ligation of the arterial duct and multiple ventricular punctures for developing hydrocephalus, followed by external ventricular drainage. After two and a half months, the patient suffered spontaneous intraparenchymal hemorrhage (both temporal lobes and right occipital lobe). The laboratory findings at that time indicated elevated, but decreasing c-reactive protein and slightly modified coagulation tests.

Genetic testing confirmed inherited thrombophilia, with a positive MTHFR A1298C heterozygous gene mutation and Antithrombin deficiency. Normal PAI-1 levels were detected. There was limited information about the family history.

### 4.14. Case No. 14—Home Onset—Right Intraventricular Hemorrhage, Post-Hemorrhagic Hydrocephalus

A 2-week-old male neonate (37 weeks, 2.80 kg) presented with post-hemorrhagic hydrocephalus of the lateral ventricles after a massive right intraventricular hemorrhage known from a head ultrasound performed at two days of life. Laboratory findings from birth showed elevated procalcitonin.

A suspicion of perinatal stroke was raised and genetic testing confirmed inherited thrombophilia, with a positive PAI-1 4G/5G promoter homozygous gene mutation, MTHFR C677T heterozygous gene mutation, and Protein S and Protein C deficiencies. Normal levels of homocysteine were detected. There was a positive family history of thrombophilia (maternal grandfather). There was no maternal history of miscarriage or other conditions throughout pregnancy.

### 4.15. Case No. 15—Postnatal Posterior Cranial Fossa Hemorrhage, Subarachnoid Hemorrhage, Thrombosis of the Right Transverse and Sigmoid Cerebral Venous Sinuses, Hydrocephalus

A 4-day-old female neonate (40 weeks, 3.10 kg) presented signs of increased intracranial pressure: projectile vomiting, bulging anterior fontanelle, and spasticity. The patient had previously developed seizures at 8 h of life, treated with phenobarbital. The first ultrasound of the brain performed at birth was normal; the second one performed at 3 days of life showed hydrocephalus of the lateral ventricles. A brain CT scan performed at 3 days of life identified right pericerebellar hemorrhage in the posterior cranial fossa, causing stenosis of the Sylvius aqueduct, hydrocephalus, and subarachnoid hemorrhage of the right temporal lobe, tentorium cerebelli, quadrigeminal, and ambiens cisterns of the brain (Figure 6). The laboratory findings from birth and 3 days of life showed leukocytosis.

The admission laboratory findings showed elevated procalcitonin. Two RT-PCR tests for SARS-CoV-2 infection were performed with negative results due to a positive maternal infection before birth. Later, a brain MRI performed at 7 days of life showed thrombosis of the right transverse and sigmoid cerebral venous sinuses and mild hydrocephalus of the lateral ventricles. A suspicion of right internal jugular vein thrombosis and elevated D-dimer tests led to the initiation of treatment with enoxaparin.

Genetic testing confirmed inherited thrombophilia, with a positive MTHFR C677T homozygous gene mutation and Protein C deficiency. Normal homocysteine and PAI-1 levels were detected. There was a positive maternal history of three miscarriages and oligohydramnios during pregnancy.

### 4.16. Case No. 16—Postnatal Ischemic Stroke

A 3-hour-old male neonate (38 weeks, 3.30 kg) presented with myelomeningocele, known since the 31st week of gestation, in order to undergo emergency surgical repair. The cardiovascular ultrasound performed on admission revealed a complex congenital cardiac malformation. During hospitalization, the patient required several invasive procedures: ventricular punctures for the persistent hydrocephalus, followed by ventriculoperitoneal shunting and surgical correction of the complex cardiac defect. Ten days later, the patient developed seizures. A head ultrasound revealed hyperechogenicity of the bilateral thalamic nuclei (possible cerebral ischemia).

Genetic testing confirmed inherited thrombophilia, with a positive MTHFR C677T heterozygous gene mutation and protein S and protein C deficiencies. Normal PAI-1 levels were detected. There was no known family history of thrombophilia or miscarriage; oligohydramnios was present during pregnancy.

### 4.17. Case No. 17—Postnatal Cerebral Thromboembolism

A 13-day-old female neonate (33 weeks, 1.50 kg) presented with a suspicion of a left pulmonary sequestration, confirmed on an admission CT scan, and severe pulmonary hypertension. Two weeks after admission, the patient developed severe bradycardia. A head ultrasound after the event indicated various hyperechoic lesions in the left temporal and parietal lobes and right insular and parietal lobes (Figure 7). The laboratory findings from that time showed anemia and elevated, but decreasing c-reactive protein. A lumbar puncture was performed, with a normal result.

Genetic testing confirmed inherited thrombophilia, with a positive MTHFR A1298C heterozygous gene mutation and Antithrombin deficiency. There was a positive maternal history of miscarriage before this pregnancy.

### 4.18. Case No. 18—Postnatal Left Intraparenchymal Hemorrhage, Thrombosis of the Left Transverse and Sigmoid Cerebral Venous Sinuses

A 3-hour-old male neonate (35 weeks, 2.59 kg), conceived by in vitro fertilization, presented with esophageal atresia with distal tracheoesophageal fistula. Two days after the admission, ligation of the tracheoesophageal fistula and gastrostomy were performed. A month after the admission, a routine head ultrasound revealed acute intraparenchymal hemorrhage in the left frontal lobe and thrombosis of the left transverse and sigmoid cerebral venous sinuses. The laboratory findings indicated mild thrombocytosis, elevated c-reactive protein, and elevated D-dimer tests.

Genetic testing confirmed inherited thrombophilia, with positive Prothrombin G20210A and MTHFR A1298C heterozygous gene mutations and Protein S and Protein C deficiencies. Normal PAI-1 levels were detected. There was no known family history of thrombophilia or miscarriages.

### 4.19. Case No. 19—In Utero Post-Hemorrhagic Hydrocephalus

A 1-day-old female neonate (38 weeks, 3.45 kg) presented with congenital hydrocephalus, known since the 24th week of gestation, from fetal ultrasound.

Genetic testing confirmed inherited thrombophilia, with positive MTHFR A1298C and MTHFR C677T heterozygous gene mutations and Antithrombin, Protein S, and Protein C deficiencies. Normal PAI-1 levels were detected. There was no known family history of thrombophilia or miscarriage, but there was a positive history of gestational hypertension during pregnancy.

### 4.20. Case No. 20—Home Onset—Intraventricular Hemorrhage

A 12-day-old female neonate (39 weeks, 2.68 kg) presented with a bulging anterior fontanelle and obtundation after being previously diagnosed with bilateral intraventricular hemorrhage at another pediatric emergency department at 9 days of life.

Genetic testing confirmed inherited thrombophilia, with positive Factor V Leiden and MTHFR C677T heterozygous gene mutations and Protein S and Protein C deficiencies. Normal PAI-1 levels were detected. There was no known family history of thrombophilia or miscarriages.

### 4.21. Case No. 21—In Utero Thrombosis of the Inferior Vena Cava, the Left Renal Vein, the Abdominal Aorta, and Both Common Iliac Arteries

A 3-day-old male neonate (39 weeks, 3.52 kg), known from birth with pulseless femoral arteries, presented with generalized cyanosis, especially to the lower limbs. Abdominal ultrasound indicated extended thrombosis of the inferior vena cava, the left renal vein, and the abdominal aorta. The abdominal CT scan also revealed thrombosis of both common iliac arteries. Elevated D-dimer tests were detected. Treatment with heparin and alteplase was initiated, followed by treatment with enoxaparin, which led only to the reduction in the left renal vein thrombosis—the others remained constant. Surgical vascular plasty was intended with success.

A suspicion of in utero thrombosis of the vessels was raised. Genetic testing confirmed inherited thrombophilia, with positive MTHFR A1298C, MTHFR C677T, PAI-1 4G/5G promoter, Factor XIII (Val34Leu), Fibrinogen 455 G>A, and GP Iib/IIIa L33P heterozygous gene mutations. Normal homocysteine levels were detected. There was no known family history of thrombophilia or miscarriage. Family screening for thrombophilia was performed afterward, with negative results for both parents.

### 4.22. Case No. 22—Perinatal Thrombosis of the Inferior Vena Cava, Both Renal Veins, and Both External Iliac Veins

A 3-day-old male neonate (36 weeks, 2.40 kg), delivered by emergency cesarean section due to fetal distress, presented with edema of the lower limbs, which rapidly progressed to the abdominal wall. Laboratory findings from birth showed severe thrombocytopenia and elevated c-reactive protein. On admission, modified coagulation tests were also identified. An abdominal ultrasound revealed thrombosis of the inferior vena cava, both renal and external iliac veins; therefore, heparin infusion was initiated. At one month of age, he developed oliguria, so continuous renal replacement therapy was initiated. However, multiple warnings of high transmembrane pressure through the CRRT circuit filter preceded the clotting of the hemodialysis circuit. Two days later, the patient died due to multiorgan failure.

Genetic testing confirmed inherited thrombophilia, with positive Prothrombin G20210A, MTHFR A1298C, and MTHFR C677T heterozygous gene mutations. There was no known family history of thrombophilia or miscarriage. At 30 weeks of gestation, the mother was admitted to the hospital for acute cystitis, with a high risk of preterm delivery.

### 4.23. Case No. 23—In Utero Thrombosis of the Left Renal Vein

A 6-day-old male neonate (36 weeks, 2.90 kg) presented with thrombosis of the left renal vein, known from the routine prenatal ultrasound, followed by an emergency C-section. The laboratory findings from birth showed positive D-dimer tests. On admission, blood tests identified neutrophilic leukocytosis and elevated D-dimer tests. Heparin infusion was initiated from birth, followed by enoxaparin. A month later, the abdominal ultrasound confirmed the dissolution of the thrombus.

A suspicion of inherited thrombophilia was raised, and genetic testing confirmed it, with a positive PAI-1 4G/5G promoter homozygous gene mutation, MTHFR C677T heterozygous gene mutation, and Protein S and Protein C deficiencies. There was no known family history of thrombophilia.

### 4.24. Case No. 24—Postnatal Thrombosis of the Mesenteric Veins and the Inferior Vena Cava

A 5-day-old male neonate (35 weeks, 1.15 kg) presented with apnea of prematurity. During hospitalization, the patient started to deteriorate clinically with feeding intolerance, abdominal tenderness and rigidity, respiratory distress, and subfebrility. Laboratory findings showed moderate anemia, highly elevated c-reactive protein, and modified coagulation tests. A cardiovascular ultrasound revealed an obstruction of the inferior vena cava, with normal blood flow of the suprahepatic veins. Abdominal radiography showed pneumoperitoneum and intestinal perforation, which required immediate surgical intervention. The resection of 25 cm of the jejunum and end-to-end jejunoileal anastomosis were performed. Thrombosis of the mesenteric veins was detected during surgery. Due to the positive family history of maternal thrombophilia, treatment with acetylsalicylic acid was initiated.

Genetic testing confirmed inherited thrombophilia, with a positive PAI-1 4G/5G promoter heterozygous gene mutation and MTHFR C677T homozygous gene mutation.

### 4.25. Case No. 25—Perinatal Ischemia of the Mesenteric Vessels with Meconium Peritonitis

A 3-day-old female neonate (37 weeks, 2.92 kg), known from birth to vomit after attempting enteral nutrition, presented with duodenal stenosis (double bubble sign on the abdominal radiography). At seven days of life, side-to-side duodenal anastomosis and the resection of the obstructing annular pancreas were performed. The surgical intervention led to the discovery of an old perforation of the right colon, which led to meconium peritonitis; thus, a right colostomy was performed. A suspicion of perinatal ischemia of the mesenteric vessels was raised (Figure 8). Elevated D-dimer tests were identified, and continuous heparin infusion was initiated, followed by treatment with enoxaparin.

Genetic testing confirmed inherited thrombophilia, positive Factor V Leiden, MTHFR A1298C and MTHFR C677T heterozygous gene mutations, PAI-1 4G/5G promoter homozygous gene mutation, and Antithrombin, Protein S, and Protein C deficiencies. Normal homocysteine levels and high methionine levels were detected. There was a positive maternal history of miscarriage.

### 4.26. Case No. 26—In Utero Perforation of the Ileum and Meconium Peritonitis

A 1-day-old female neonate (35 weeks, 2.25 kg) presented with meconium peritonitis, known since the 25th week of gestation, in order to undergo emergency surgical repair. During hospitalization, she presented multiple episodes of abnormal sk—coloration—vasodilation of the hands, forearms, thorax, and upper abdominal wall and vasoconstriction of the head, neck, arms, shoulders, lower abdominal wall, and lower limbs. Elevated D-dimer tests were detected. Continuous heparin infusion was initiated, but was stopped after five days due to the suspicion of intraabdominal bleeding. The patient died at 15 days of life.

Genetic testing confirmed inherited thrombophilia, with positive MTHFR A1298C heterozygous gene mutations and Protein S and Protein C deficiencies. Normal PAI-1 levels were detected. There was a positive maternal history of miscarriage.

### 4.27. Case No. 27—In Utero Thrombosis of the Portal Vein

A 5-day-old female neonate (38 weeks, 2.99 kg) presented with esophageal atresia and distal tracheoesophageal fistula, associated with coarctation of the aorta, which was later surgically corrected. During hospitalization, a routine abdominal ultrasound revealed the absence of the portal vein trunk and its branching inside the liver. A suspicion of in utero thrombosis of the portal vein with subsequent collateral development was raised.

Genetic testing confirmed inherited thrombophilia, with a positive MTHFR A1298C heterozygous gene mutation and Protein S and Protein C deficiencies. A mild elevation of PAI-1 was also detected. There was no known family history of thrombophilia or miscarriages. The mother had a SARS-CoV-2 infection at 22 weeks of gestation and an HPV infection during pregnancy.

### 4.28. Case No. 28—Postnatal Thrombosis of the Inferior Vena Cava

A 2-day-old male neonate (38–39 weeks, 3.5 kg) presented with transposition of the great vessels and renal failure. The patient’s severe condition required the immediate initiation of continuous renal replacement therapy, with continuous heparin infusion for the hemodialysis device. Two days after the surgical correction of the cardiac malformation, a routine abdominal ultrasound identified thrombosis of the inferior vena cava (Figure 9). The laboratory findings showed mild thrombocytopenia, elevated, but decreasing c-reactive protein, and modified coagulation tests (due to ongoing anticoagulation treatment).

During hospitalization, thrombi were noticed in the deaeration chamber, and multiple warnings of filter clotting were received from the CRRT device, leading to several heparin bolus administrations and multiple circuit replacements. During the last four days of CRRT, the patient required three circuit changes, taking into account that a circuit usually needs replacement after a period of minimum of 72 h. The patient’s condition continued to worsen, with a negative outcome at one month of life.

Genetic testing confirmed inherited thrombophilia, with positive MTHFR A1298C and MTHFR C677T heterozygous gene mutations. Normal PAI-1 levels were detected. There was no known family history of thrombophilia or miscarriages.

### 4.29. Case No. 29—Postnatal Thrombosis of the Inferior Vena Cava, Both Renal Veins, the Left Common Iliac Vein

A 3-day-old male neonate (38 weeks, 3 kg) presented with sepsis and acute renal failure, which developed at 48 h of life. The laboratory findings from birth showed mild thrombocytopenia and elevated c-reactive protein. On admission, normal coagulation tests and D-dimer tests were detected. An abdominal ultrasound revealed thrombosis of the inferior vena cava, both renal veins, and the left common iliac vein (Figure 10). Continuous renal replacement therapy was initiated, with continuous heparin infusion for the hemodialysis device. Treatment with bivalirudin was associated for two days, and then switched to alteplase. The D-dimer values increased to undetectable levels.

Even though revascularization of the right renal vein was detected on ultrasound after ten days of continuous infusion with alteplase, anuria persisted. Multiple warnings of filter clotting and high transmembrane pressure through the filter were received, which led to several heparin bolus administrations and multiple replacements of the CRRT circuit. The patient died at 20 days of life due to multiorgan failure.

Genetic testing confirmed inherited thrombophilia, with a positive MTHFR A1298C heterozygous gene mutation and Antithrombin, Protein S, and Protein C deficiencies. A moderate elevation of PAI-1 was also detected. There was a positive family history of maternal thrombophilia, one maternal miscarriage, and an older brother known with perinatal stroke.

### 4.30. Case No. 30—Postnatal Severe Skin Ischemia and Necrosis

An 8-week-old female infant (34 weeks, 1.72 kg) presented with a complex congenital cardiac malformation for an elective surgical intervention. After admission, generalized seizures occurred, and cutis marmorata appeared during long periods of agitation. The patient started slowly developing cyanosis of the left upper limb fingers, which progressed to extended necrosis of the left upper limb. New necrosis sites appeared on the right ear, left cheek, inferior lip, tongue, and chin (Figure 11). During this period, the laboratory findings showed elevated c-reactive protein and modified coagulation tests, with D-dimer tests normal initially and elevated afterward. Treatment with pentoxifylline, enoxaparin, and alprostadil was initiated, with no effect.

Genetic testing confirmed inherited thrombophilia, with positive PAI-1 4G/5G promoter, MTHFR A1298C, and MTHFR C677T heterozygous gene mutations and Antithrombin, Protein S, and Protein C deficiencies. There was no known family history of thrombophilia. The mother had a previous therapeutic abortion due to a prenatal diagnosis of a complex congenital cardiac malformation.

### 4.31. Case No. 31—Postnatal Severe Skin Ischemia and Necrosis

A 1-day-old male neonate (36 weeks, 1.94 kg) presented with VACTERL disorder (anal atresia, complex congenital cardiac malformation, horseshoe kidney, hypospadias, and right radial aplasia). We performed Whole Exome Sequencing, which identified CHARGE syndrome. During hospitalization, the patient had multiple septic episodes and underwent several invasive procedures: colostomy, pulmonary artery banding, chylothorax drainage, several ventricular punctures for hydrocephalus, and the placement of an external ventricular drain.

The patient started to develop frequent episodes of metabolic acidosis, bradycardia, desaturation, and cyanosis of the right hemiface, thorax, left upper limb, and right lower limb, which slowly turned to ischemia. Necrosis of the left hand occurred. Treatment with pentoxifylline, alprostadil, and heparin was initiated without effect, leading to amputation of the left hand in order to stop the advancing necrosis (Figure 12).

Several months later, new ischemia sites appeared on the right hand. Treatment with heparin was reinitiated, followed by enoxaparin, with no effect. Necrosis of the right-hand phalanges occurred. Continuous infusion with alprostadil was started again and daily bandages with Betadine were placed until spontaneous amputation of the phalanges occurred.

Genetic testing confirmed inherited thrombophilia, with positive PAI-1 4G/5G promoter, MTHFR A1298C, and MTHFR C677T heterozygous gene mutations. There was no known family history of thrombophilia, but maternal screening for thrombophilia was performed afterward with a positive result. The mother reported a miscarriage after this pregnancy.

### 4.32. Case No. 32—Postnatal Thrombosis of the Left Femoral Artery

A 1-day-old male neonate (33 weeks, 1.55 kg) presented with a suspicion of esophageal atresia and distal tracheoesophageal fistula, which were confirmed on admission. During hospitalization, the patient underwent several invasive procedures: ligation of the distal tracheoesophageal fistula, surgical ligation of the patent arterial duct, and several ventricular punctures for developing hydrocephalus. Meanwhile, the patient developed an occlusion of the left femoral artery. Three days prior to this event, the central arterial catheter on the right femoral artery was removed, while a central venous catheter was placed on the left femoral vein. An elevated, but slightly decreasing c-reactive protein was identified. Three weeks later, the thrombus could no longer be identified. The patient died several months after discharge due to a respiratory infection.

Genetic testing confirmed inherited thrombophilia, with positive MTHFR A1298C heterozygous gene mutation and Antithrombin, Protein S, and Protein C deficiencies. Normal PAI-1 levels were detected. There was no known family history of thrombophilia or miscarriages, but there was a positive history of gestational hypertension during pregnancy.

### 4.33. Case No. 33—In Utero Right Upper Limb Ischemia and Necrosis, Perinatal Thrombosis of Inferior Vena Cava

A 1-day-old male neonate (38 weeks, 3.13 kg) presented with ischemia and necrosis of the right upper limb (Figure 13A,B), treated with pentoxifylline and continuous heparin infusion. On admission, leukocytosis, elevated c-reactive protein, modified coagulation tests (due to ongoing anticoagulation treatment), and a mild elevation of the D-dimer tests were detected. Due to the positive maternal history of SARS-CoV-2 infection during the first trimester of pregnancy, an RT-PCR test for SARS-CoV-2 infection was performed, with a negative result. An abdominal ultrasound revealed an old thrombus in the inferior vena cava (Figure 13C). During hospitalization, treatment with heparin and pentoxifylline was continued, to which bivalirudin was added.

Genetic testing confirmed inherited thrombophilia, with positive MTHFR A1298C and MTHFR C677T heterozygous gene mutations and Antithrombin deficiency. A mild elevation of PAI-1 levels was detected. Subsequently, the patient received thiamin and pyridoxine supplementation. There was no known family history of thrombophilia or miscarriages.

### 4.34. Case No. 34—Postnatal Thrombosis of the Right Axillary Artery

A 10-day-old female neonate (36 weeks, 2.37 kg), known before birth with Dandy–Walker Syndrome, presented with hydrocephalus and severe coarctation of the aorta. Two central arterial lines were placed—in the right brachial and the left femoral arteries—and the aortic defect was corrected. One day later, hematoma of the right arm and ecchymosis near the insertion site of the brachial arterial catheter were noticed. Vascular ultrasound could not reveal any blood flow on the axillary or the proximal brachial arteries, leading to the removal of the central arterial catheter (Figure 14). Continuous heparin infusion was initiated, followed by treatment with enoxaparin. Later, blood flow was reestablished on both axillary and brachial arteries.

Genetic testing confirmed inherited thrombophilia, with a positive MTHFR C677T homozygous gene mutation and Antithrombin, Protein S, and Protein C deficiencies. Normal PAI-1 levels were detected. There was a positive family history of maternal thrombophilia (MTHFR A1298C, MTHFR C677T, PAI-1 4G/5G promoter, and Factor XIII (Val34Leu) heterozygous gene mutations and Protein S deficiency), treated with enoxaparin and acetylsalicylic acid during pregnancy. There were no maternal miscarriages.

### 4.35. Case No. 35—Home Onset—Thrombosis of the Right Pulmonary Artery and Intracardiac Thrombosis

A 19-day-old male neonate (35–36 weeks, 2.20 kg) presented with thrombosis of the right pulmonary artery diagnosed on the same day at another pediatric emergency department, where treatment with continuous heparin infusion was initiated. The admission laboratory findings showed elevated c-reactive protein, modified coagulation tests (ongoing anticoagulation treatment), and elevated D-dimer tests. The cardiac ultrasound identified thrombi in the right atrium and ventricle (Figure 15). Therapy with heparin was continued, to which alteplase was added. After four days of anticoagulation and fibrinolytic therapy, the thrombi could no longer be noticed on ultrasound or CT scan.

Genetic testing confirmed inherited thrombophilia, with a positive MTHFR C677T homozygous gene mutation and Antithrombin, Protein S, and Protein C deficiencies. A mild elevation of PAI-1 levels was detected. There was limited information about the family history.

### 4.36. Case No. 36—Postnatal Intracardiac Thrombosis

A 7-week-old male infant (36 weeks, 2.50 kg), the first twin from a twin pregnancy, presented with a complex congenital cardiac malformation for an elective surgical procedure (Blalock–Taussig shunt). Heparin infusion for the shunt was started immediately after the intervention and anticoagulation was continued with enoxaparin due to good clinical condition. Then, 18 days after the procedure, a routine cardiovascular ultrasound revealed a thrombus in the left ventricle (Figure 16). The laboratory studies showed leukocytosis, highly elevated c-reactive protein, and modified coagulation tests (due to anticoagulant treatment). Continuous heparin infusion was reinitiated, but multiple intracardiac thrombi were detected during the following days. Treatment with alteplase was also started, but the patient’s condition continued deteriorating, leading to a negative outcome due to cardiac failure.

Genetic testing confirmed inherited thrombophilia, with positive PAI-1 4G/5G promoter, MTHFR A1298C homozygous gene mutations, and Antithrombin and Protein C deficiencies. There was no known family history of thrombophilia. The mother had a previous therapeutic abortion due to a prenatal diagnosis of a complex congenital cardiac malformation.

### 4.37. Case No. 37—Postnatal Intracardiac Thrombosis

A 1-day-old male neonate (unknown gestational age, 3.00 kg) presented with a complex congenital cardiac malformation for an elective surgical procedure. Admission laboratory findings showed elevated c-reactive protein, procalcitonin, and slightly modified coagulation tests. A routine cardiovascular ultrasound showed multiple thrombi in the left ventricle (Figure 17). Continuous heparin infusion was initiated. The patient’s condition continued to deteriorate, developing pneumothorax at one month of life, leading to a negative outcome.

Genetic testing confirmed inherited thrombophilia, with positive Prothrombin G20210A and MTHFR A1298C heterozygous gene mutations and Antithrombin, Protein S, and Protein C deficiencies. A moderate elevation of PAI-1 levels was also detected. There was a positive maternal history of miscarriage.

### 4.38. Case No. 38—Postnatal Chylothorax

A 2-day-old male neonate (32 weeks, 1.95 kg) presented with a posterior urethral valve and bilateral renal urinomas. During hospitalization, the patient’s condition was extremely severe, requiring drainage of the urinomas, abdominal paracentesis, and several reintubations. Five weeks after the admission, left chylothorax was noticed. Even though no venous thrombosis was detected, the central venous catheter from the left internal jugular vein was removed. The laboratory findings from that time indicated leukocytosis, elevated c-reactive protein, and modified coagulation tests. Elevated D-dimer tests were detected. The patient died at six weeks of life due to cardiopulmonary arrest.

Genetic testing confirmed inherited thrombophilia, with positive Factor V Leiden and MTHFR C677T heterozygous gene mutations. Normal PAI-1 levels were detected. There was limited information available regarding the patient’s family history.

### 4.39. Case No. 39—Postnatal Thrombosis of Both Internal Jugular Veins, Both Brachiocephalic Veins, the Superior Vena Cava, the Inferior Vena Cava, Both Common Iliac Veins, Both External Iliac Veins, and Both Femoral Veins

A 6-week-old male neonate (29 weeks, 1.23 kg), the first twin from a twin pregnancy, known with cardiopulmonary arrest with prolonged cardiopulmonary resuscitation, presented with severe bronchopulmonary dysplasia. The admission laboratory findings showed leukocytosis, modified coagulation tests, elevated D-dimer tests, and sepsis with Pseudomonas aeruginosa. During hospitalization, the patient suffered multiple fractures of the humerus, femur, tibia, and costal ribs.

A cardiovascular ultrasound revealed thrombosis of the right internal jugular and brachiocephalic veins. Therefore, the central venous catheter from the right internal jugular vein was removed, and two other central lines were inserted on the right femoral vein and on the left internal jugular vein. Despite initiating anticoagulant treatment with heparin, followed by bivalirudin and fibrinolytic agents like alteplase, the CT angiography indicated an extension of the thrombus to the right atrium (Figure 18). A thrombectomy and pericardial patch venoplasty were performed at four months of age. Despite the double anticoagulation treatment with heparin and bivalirudin, the superior vena cava syndrome clinically persisted, and thrombosis of the left internal jugular and brachiocephalic veins also occurred. Thrombosis of the inferior vena cava, both common iliac veins, external iliac veins, and femoral veins were diagnosed in turn. The patient died at 14 months of life due to cardiopulmonary arrest.

A suspicion of inherited thrombophilia was raised, and genetic testing confirmed it, with a positive MTHFR C677T homozygous gene mutation and antithrombin deficiency. Normal PAI-1 levels were detected. There was a positive family history of thrombosis—the second twin from the delivery, who died at six weeks of life, presented cerebral venous sinus thrombosis on the brain ultrasound from admission.

### 4.40. Case No. 40—Postnatal Thrombosis of Both Internal Jugular Veins, Superior and Inferior Venae Cavae, and Both Femoral Veins

A 3-hour-old female neonate (38 weeks, 3.00 kg) presented with gastroschisis, known since the 18th week of gestation, to undergo emergency surgical repair of the anterior abdominal defect. A second surgical intervention for inspecting the abdominal organs was required during hospitalization, and an ileostomy was performed. Two days later, thrombosis of the right internal jugular vein was noticed, which led to the removal of the catheter and the insertion of another central line on the left internal jugular vein. Laboratory findings showed elevated c-reactive protein. The patient developed bilateral chylothorax after six days. The catheter from the left internal jugular vein was also removed, and another one was placed on the left femoral vein. One day later, a cardiac ultrasound indicated thrombosis of the superior vena cava. Continuous heparin infusion was initiated, followed by treatment with enoxaparin. No other thrombi were formed during that time. However, during hospitalization, new thrombus formation was noticed on the left internal jugular vein, both femoral veins, and the inferior vena cava.

Genetic testing confirmed inherited thrombophilia, with positive Prothrombin G20210A and MTHFR C677T heterozygous gene mutations and Protein C deficiency. There was no known family history of thrombophilia or miscarriage.

## 5. Discussion

This case series illustrates multiple sites for the thrombotic events, occurring mostly in the vessels of the brain, but also vessels from other parts of the body, such as the abdomen, limbs, and thorax. As mentioned above, 28 patients had a positive outcome, being discharged from our unit. For six of them, complete revascularization was possible, but one case had a negative outcome due to a respiratory infection after discharge. For the other discharged patients, symptomatic treatment was possible, like resection of the small intestine, colostomy, and external ventricular drains followed by ventriculoperitoneal shunts or daily bandages of the skin lesions. As for the deceased patients, the thrombotic events contributed to some extent to their negative outcome, but in five cases, the appearance of the thrombi accelerated the moment of death. One more patient with incomplete revascularization and negative outcome during hospitalization should also be mentioned.

Some of the cases we presented are critically ill neonates admitted in our NICU who developed thrombosis during the acute stages of their diseases after long-term hospitalization or multiple invasive interventions, as well as repeated periods of increased inflammation, oxidative stress, and vascular endotheliitis. However, the rest of the patients had not been previously exposed to any stress, having normal blood studies, with home or in utero onset of serious thrombotic events, which later worsened the quality of their lives and of their parents.

Considering that 35 of the patients presented in this case series had positive genetic results for MTHFR variants accompanied by several risk factors, we cannot establish exactly the causal effect between MTHFR mutations and the development of thrombosis. In five cases, additional pro-thrombotic conditions were not identified, highlighting the essential contribution of MTHFR to thrombosis, but not the causation. Moreover, the other congenital thrombophilia mutations identified in 92.5% of the cases have a significant impact on the pathogenesis of thrombosis. Thus, the degree to which MTHFR mutations influence thrombotic conditions cannot be precisely determined.

Healthcare providers, especially neonatologists and gynecologists, should raise a high index of suspicion facing a positive MTHFR genetic result. In 11 cases from our series, thrombosis occurred in utero. Currently, MTHFR variants are not recommended for routine testing during pregnancy and anticoagulant therapy is only recommended in mothers with other inherited thrombophilia conditions [35]. As pregnancy alone is considered a hypercoagulable state and MTHFR contributes to it, we advise other physicians to be aware of the high risk associated with a positive genetic result.

Furthermore, as NICU patients could easily develop (sub)febrile states, neonatologists should consider the risk of thrombosis, especially for C677T homozygotes [5], as the MTHFR enzyme is thermolabile and reduces its activity to less than half at temperatures higher than 37 °C compared to a normal enzyme.

Even in asymptomatic adults positive for homozygous or double heterozygosity containing factor V Leiden, prophylactic anticoagulation is not recommended, and significant risks surpass the benefits, like bleeding, bone loss, and thrombocytopenia [36]. However, in high-risk situations such as trauma, major surgery, prolonged immobilization, pregnancy, and the first six weeks postpartum, prophylactic anticoagulation may be considered for the asymptomatic adult with a thrombophilic defect [37].

There is a lack of information in the literature regarding the preventive management of thrombosis. There are no recommendations concerning the treatment of asymptomatic neonates. According to current recommendations [32,33,34], prevention is the main objective of the treatment, whereas anticoagulation and, rarely, antithrombotic therapy are administered in already developed and severe thrombosis cases. However, children with numerous pro-thrombotic conditions can receive prophylactic anticoagulation therapy during hospitalization [38]. We propose the extension of this recommendation to neonates with significant risk factors, including MTHFR mutations.

Compared to adults [36], indefinite anticoagulation following a thrombosis event is not routinely recommended in the pediatric population, as the risks surpass the benefits [38]. We recommend close monitoring of the discharged patient who carries an MTHFR mutation, especially in the presence of a primary thrombotic event. Identifying and preventing further risk factors should be of great importance to the physicians and families of these patients.

Nonetheless, as we revealed in our cases, initiating treatment after the first thrombotic event may be too late, greatly impacting the condition of critically ill neonates. As indicated by the available guidelines, treatment with anticoagulants or thrombolytics was initiated after the thrombosis occurred and became symptomatic. Only two patients received anticoagulants before the thrombosis occurred because of the presence of the hemodialysis device and the Blalock–Taussig shunt. Both developed thrombosis events, nevertheless.

There are no known medications for thrombosis other than anticoagulant or antithrombotic therapy.

In critically ill neonates in intensive care units, seriously considering screening for MTHFR mutations becomes imperative to prevent thrombosis. However, our hospital’s current testing process for inherited thrombophilia poses challenges. The laboratory requirements for multiple blood samples are inconvenient for our patients, especially considering that, upon admission, various blood samples are also collected. Additionally, some patients may already present with anemia, further complicating the process. Consequently, the collection of blood samples for thrombophilia testing often occurs over separate days rather than simultaneously, and the genetic results usually arrive too late. Therefore, there is a pressing need for a new testing approach.

We propose implementing a screening procedure for all patients upon admission, utilizing a simple diagnostic dipstick method. This streamlined method would provide immediate results and facilitate the early identification of positive cases. Subsequently, positive test results could undergo further genetic analysis to identify the specific mutation associated with MTHFR. This approach would be highly beneficial in optimizing the management of thrombotic risks in our patient population. We strongly recommend screening for MTHFR variants in the NICU to identify them as risk factors for thrombosis and correlate them with other risk factors that may appear during further hospitalization.

The limitations of our study are indeed the association of other important risk factors for thrombosis in our patients, including various congenital thrombophilia mutations. As we only tested for MTHFR mutation in patients from our unit who presented with thrombosis, leading to a positive result, we cannot state the actual frequency of this mutation in all NICU patients or the general neonatal population, nor its association with other medical conditions.

## 6. Conclusions

The particularity of this case series is that every patient in our unit who developed thrombi had a positive genetic result for MTHFR mutations. MTHFR mutations should be regarded as thrombotic risk factors for our critically ill patients, and screening for MTHFR mutations should be commenced in every admitted patient to intensive care units. Thus, the prevention of thrombi could be performed. It is important as, with every positive genetic result of MTHFR in patients who had not developed thrombosis, every measure must be made to avoid other pro-thrombotic risk factors. Moreover, treatment should be started in patients with positive genetic results who experience invasive medical interventions.

## Figures and Tables

**Figure 1 jcm-12-04856-f001:**
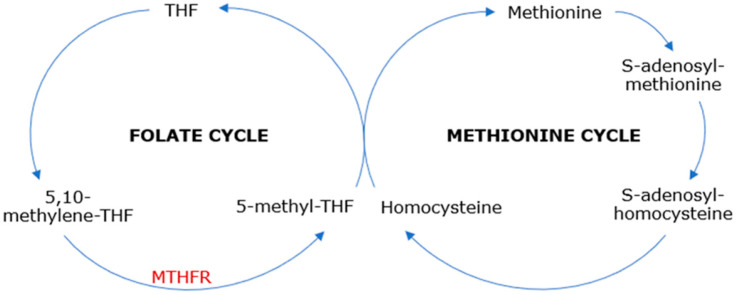
Folate and methionine cycles. THF—tetrahydrofolate, MTHFR—methylenetetrahydrofolate reductase.

**Figure 2 jcm-12-04856-f002:**
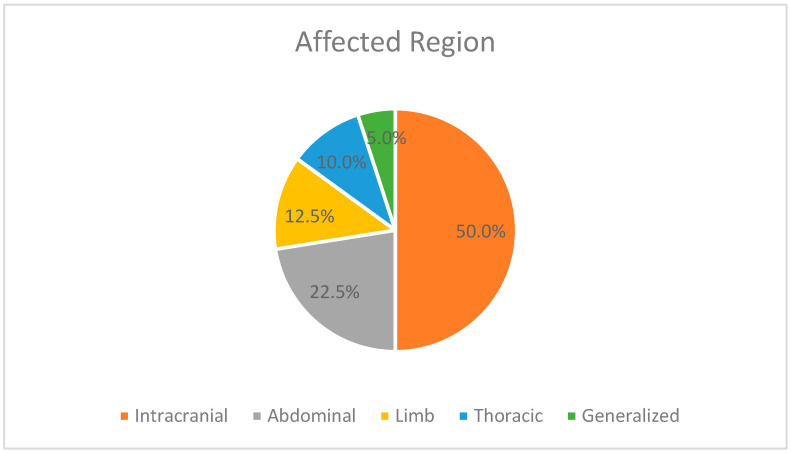
Regions affected by thrombosis.

**Figure 3 jcm-12-04856-f003:**
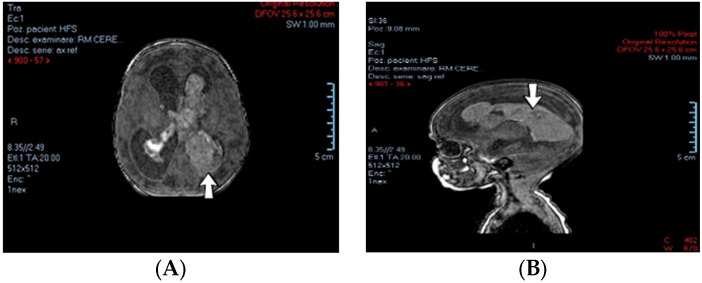
Case no. 1. Left intraventricular hemorrhage on brain MRI—transverse (**A**) and sagittal (**B**) views.

**Figure 4 jcm-12-04856-f004:**
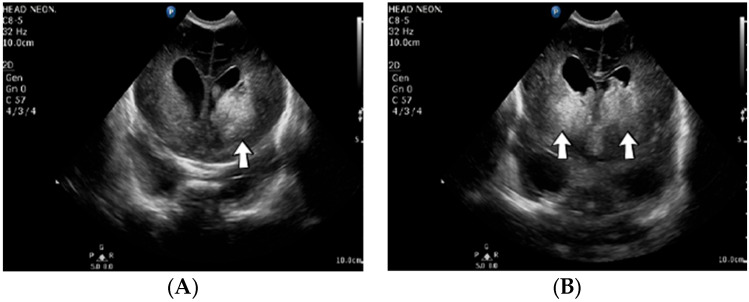
Case no. 4. Left intraparenchymal hemorrhage (**A**) from admission and bilateral basal ganglia hemorrhage three days after the admission (**B**) on brain ultrasound.

**Figure 5 jcm-12-04856-f005:**
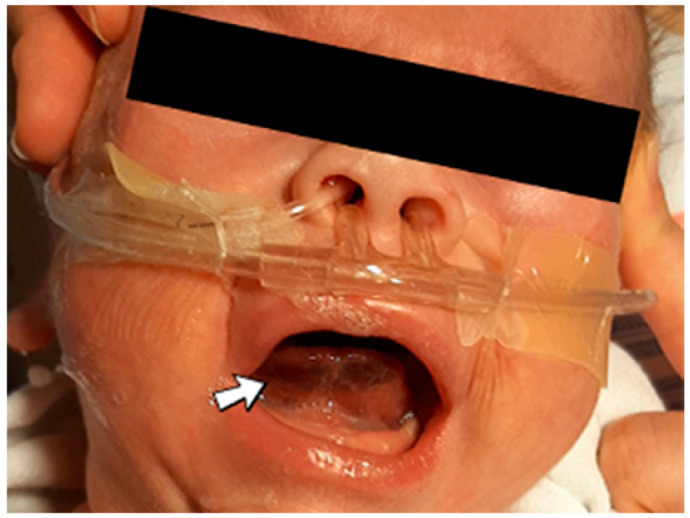
Case no. 10. Sublingual hematoma.

**Figure 6 jcm-12-04856-f006:**
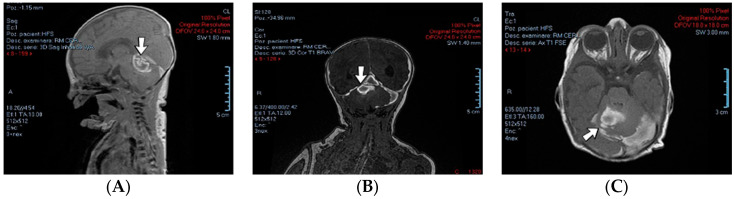
Case no. 15. Posterior cranial fossa hemorrhage on brain MRI—sagittal (**A**), frontal (**B**), and transverse (**C**) views.

**Figure 7 jcm-12-04856-f007:**
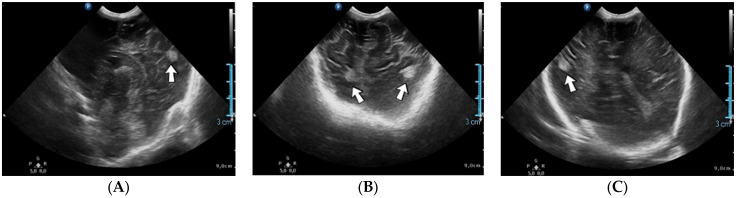
Cerebral thromboembolism on brain ultrasound: parietal lobes (**A**–**C**).

**Figure 8 jcm-12-04856-f008:**
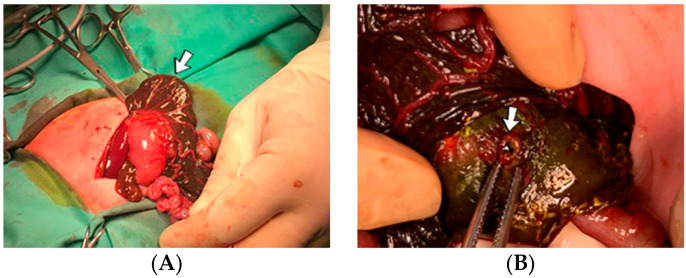
Case no. 25. During the surgery, the mesenteric vessels were noticed (**A**), and an old perforation (**B**) of the right colon was discovered, leading to meconium peritonitis.

**Figure 9 jcm-12-04856-f009:**
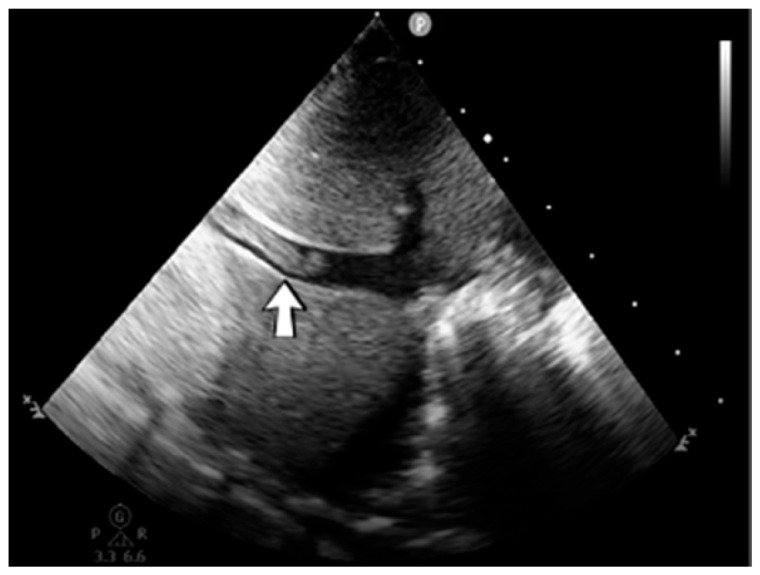
A thrombus inside of the inferior vena cava was noticed on abdominal ultrasound.

**Figure 10 jcm-12-04856-f010:**
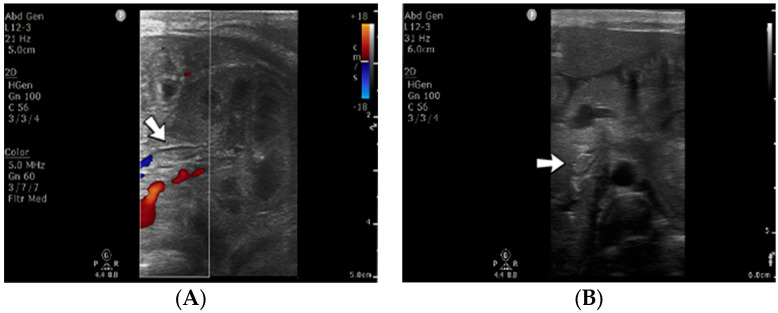
Thrombosis of the left renal vein (**A**) and thrombosis of the inferior vena cava (**B**) were noticed on abdominal ultrasound. No Doppler signal was identified on the renal vein versus the renal artery (**A**).

**Figure 11 jcm-12-04856-f011:**
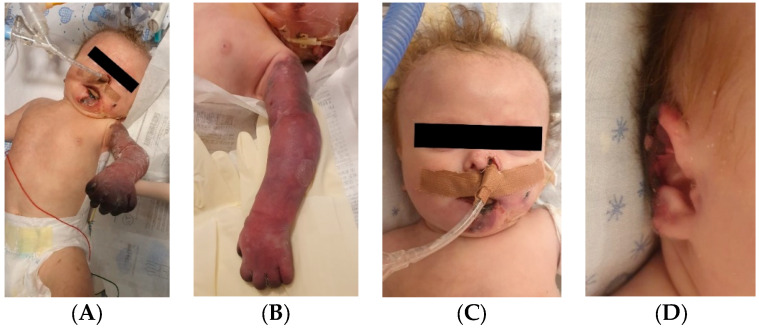
Case no. 30. Clinical presentation of the skin ischemia and necrosis: left upper limb (**A**,**B**), left cheek, inferior lip, tongue and chin (**C**), right ear (**D**).

**Figure 12 jcm-12-04856-f012:**
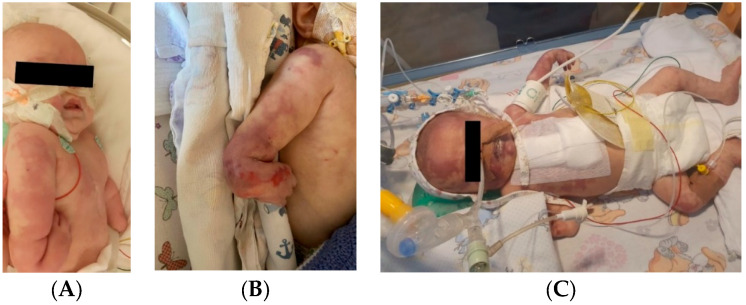
Clinical presentation of the skin ischemia and necrosis: thorax and right upper limb (**A**), right upper limb (**B**), right hemiface, left upper limb, and right lower limb (**C**).

**Figure 13 jcm-12-04856-f013:**
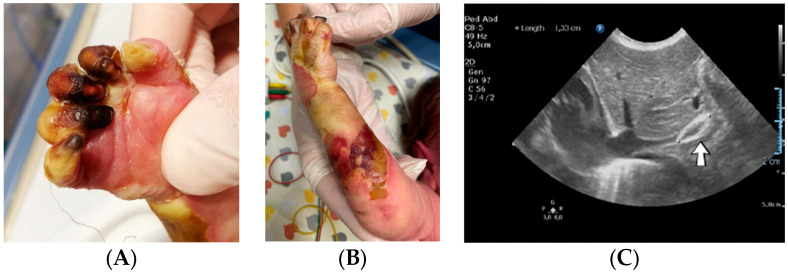
Clinical presentation of the skin ischemia and necrosis: fingers of the right upper limb (**A**), right forearm (**B**). An old inferior vena cava thrombosis was shown on abdominal ultrasound (**C**).

**Figure 14 jcm-12-04856-f014:**
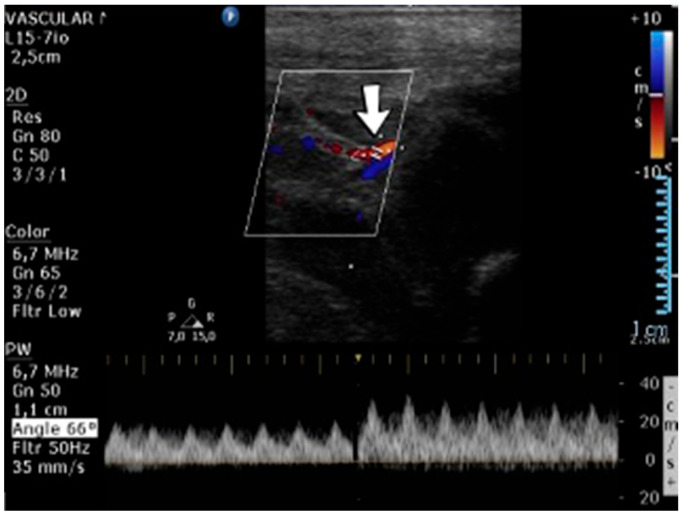
No blood flow on the axillary or the proximal brachial arteries on vascular ultrasound.

**Figure 15 jcm-12-04856-f015:**
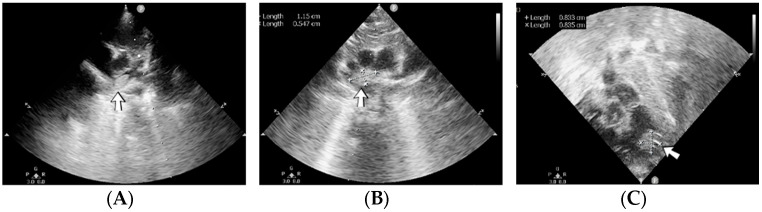
Intracardiac thrombosis was identified on cardiac ultrasound: in the right pulmonary artery (**A**), the right atrium (**B**), and the right ventricle (**C**).

**Figure 16 jcm-12-04856-f016:**
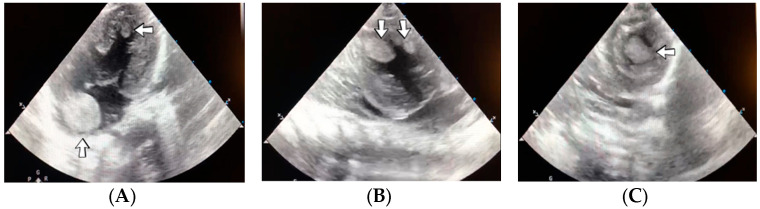
Intracardiac thrombosis identified on cardiac ultrasound: left atrium and ventricle (**A**), left ventricle (**B**), apex (**C**).

**Figure 17 jcm-12-04856-f017:**
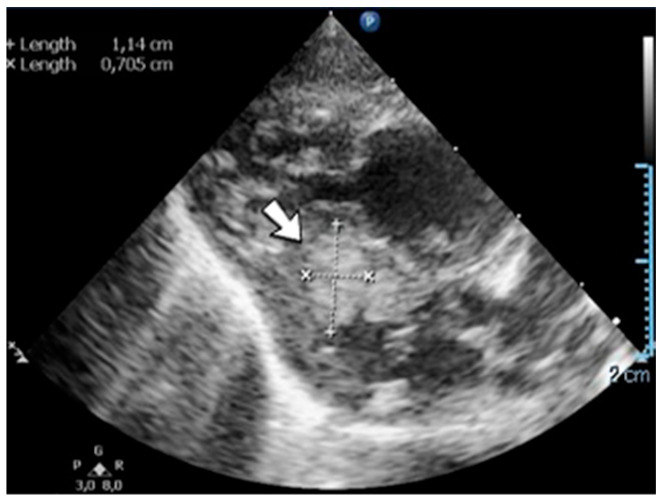
Intracardiac thrombosis was identified on cardiac ultrasound in the left ventricle.

**Figure 18 jcm-12-04856-f018:**
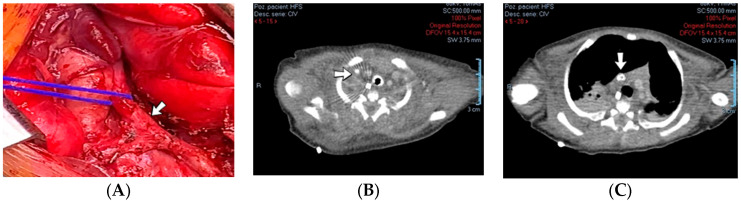
Case no. 39. During surgery, a fibrous band (**A**) was noticed between the right internal jugular and brachiocephalic veins. Thrombosis of the right internal jugular vein (**B**) and the superior vena cava (**C**) was identified on Angio-CT.

**Table 1 jcm-12-04856-t001:** Summary results of the patients included in this study.

	Results
Number of cases	40
Positive family history	13
Family screening	5
In utero thrombosis	11
Home onset	4
Repeated thrombosis event	8
Complete revascularization	6

**Table 2 jcm-12-04856-t002:** General characteristics of the patient population including the diagnosis and the genetic result.

#	Affected Region	Postpartum Diagnosis	Pro-Thrombotic Conditions	Genetic Result	Outcome
1	Intracranial	Left intraventricular and intraparenchymal hemorrhage	None	MTHFR A1298C—heterozygous Fibrinogen 455 G>A—heterozygous Protein C deficiency	Discharged
2	Intracranial	Right intraventricular hemorrhage Post-hemorrhagic hydrocephalus	Pregnancy	Factor V Leiden—heterozygous Factor V A4070G (HR2 haplotype)—heterozygous MTHFR A1298C—heterozygous PAI-1 4G/5G promoter—heterozygous Factor XIII (Val34Leu)—heterozygous Fibrinogen 455 G>A—heterozygous	Discharged
3	Intracranial	Perinatal stroke	Maternal Thrombophilia	MTHFR A1298C—heterozygous MTHFR C677T—heterozygous	Discharged
4	Intracranial	Intraparenchymal and intraventricular hemorrhage	Maternal Thrombophilia	MTHFR A1298C—heterozygous MTHFR C677T—heterozygous Antithrombin deficiency Protein S deficiency Protein C deficiency	Discharged
5	Intracranial	Right intraventricular hemorrhage Post-hemorrhagic hydrocephalus	Maternal Thrombophilia Family history of thrombophilia Pregnancy	MTHFR A1298C—homozygous Protein C deficiency	Discharged
6	Intracranial	Post-hemorrhagic hydrocephalus	Pregnancy	MTHFR A1298C—heterozygous MTHFR C677T—heterozygous Antithrombin deficiency Protein S deficiency Protein C deficiency	Discharged
7	Intracranial	Cerebellar hemorrhage Post-hemorrhagic hydrocephalus Intraparenchymal hemorrhage	Maternal Thrombophilia Pregnancy	Factor V Leiden—heterozygous MTHFR C677T—homozygous Protein S deficiency Protein C deficiency	Discharged
8	Intracranial	Intraparenchymal hemorrhage	Maternal Thrombophilia	PAI-1 4G/5G promoter—homozygous MTHFR C677T—homozygous Antithrombin deficiency Protein S deficiency Protein C deficiency	Discharged
9	Intracranial	Intraventricular hemorrhage Nonobstructive hydrocephalus	None	MTHFR A1298C—heterozygous MTHFR C677T—heterozygous Antithrombin deficiency Protein C deficiency	Discharged
10	Intracranial	Right intraventricular and intraparenchymal hemorrhage Sublingual hematoma	Major surgery	Prothrombin G20210A—heterozygous PAI-1 4G/5G promoter—heterozygous MTHFR A1298C—heterozygous MTHFR C677T—heterozygous Antithrombin deficiency Protein S deficiency Protein C deficiency	Discharged
11	Intracranial	Post-hemorrhagic hydrocephalus	Pregnancy	PAI-1 4G/5G promoter—homozygous MTHFR A1298C—heterozygous MTHFR C677T—heterozygous Protein S deficiency Protein C deficiency	Discharged
12	Intracranial	Cerebral venous sinus thrombosis	Acute condition Major surgery	Factor V Leiden—heterozygous Activated Protein C Resistance V MTHFR A1298C—heterozygous Antithrombin deficiency Protein S deficiency Protein C deficiency	Discharged
13	Intracranial	Intraparenchymal hemorrhage	Major Surgeries Severe chronic illness	MTHFR A1298C—heterozygous Antithrombin deficiency	Hospitalized
14	Intracranial	Right intraventricular hemorrhage Post-hemorrhagic hydrocephalus	Family history of thrombophilia	PAI-1 4G/5G promoter—homozygous MTHFR C677T—heterozygous Protein S deficiency Protein C deficiency	Discharged
15	Intracranial	Posterior cranial fossa and subarachnoid hemorrhage Cerebral venous sinus thrombosis Hydrocephalus	Maternal SARS-CoV-2 infection Maternal miscarriage	MTHFR C677T—homozygous Protein C deficiency	Discharged
16	Intracranial	Ischemic stroke	Acute conditions Major surgeries	MTHFR C677T—heterozygous Protein S deficiency Protein C deficiency	Discharged
17	Intracranial	Cerebral thromboembolism	Acute conditions Maternal miscarriage	MTHFR A1298C—heterozygous Antithrombin deficiency	Deceased
18	Intracranial	Left intraparenchymal hemorrhage Cerebral venous sinus thrombosis	Acute condition Major surgery	Prothrombin G20210A—heterozygous MTHFR A1298C—heterozygous Protein S deficiency Protein C deficiency	Discharged
19	Intracranial	Post-hemorrhagic hydrocephalus	Pregnancy	MTHFR A1298C—heterozygous MTHFR C677T—heterozygous Antithrombin deficiency Protein S deficiency Protein C deficiency	Discharged
20	Intracranial	Intraventricular hemorrhage	None	Factor V Leiden—heterozygous MTHFR C677T—heterozygous Protein S deficiency Protein C deficiency	Discharged
21	Abdominal	Thrombosis of the inferior vena cava, the left renal vein, the abdominal aorta, and both common iliac arteries	Pregnancy	MTHFR A1298C—heterozygous MTHFR C677T—heterozygous PAI-1 4G/5G promoter—heterozygous Factor XIII (Val34Leu)—heterozygous Fibrinogen 455 G>A—heterozygous GP IIb/IIIa L33P—heterozygous	Discharged
22	Abdominal	Thrombosis of the inferior vena cava, both renal veins, and both external iliac veins	None	Prothrombin G20210A—heterozygous MTHFR A1298C—heterozygous MTHFR C677T—heterozygous	Deceased
23	Abdominal	Thrombosis of the left renal vein	Pregnancy	PAI-1 4G/5G promoter—homozygous MTHFR C677T—heterozygous Protein S deficiency Protein C deficiency	Discharged
24	Abdominal	Thrombosis of the mesenteric veins and the inferior vena cava	Sepsis Maternal Thrombophilia	PAI-1 4G/5G promoter—heterozygous MTHFR C677T—homozygous	Discharged
25	Abdominal	Perinatal ischemia of the mesenteric vessels with meconium peritonitis	Acute condition Maternal miscarriage	PAI-1 4G/5G promoter—homozygous Factor V Leiden—heterozygous MTHFR A1298C—heterozygous MTHFR C677T—heterozygous Antithrombin deficiency Protein S deficiency Protein C deficiency	Discharged
26	Abdominal	Perforation of the ileum and meconium peritonitis	Maternal miscarriage Pregnancy	MTHFR A1298C—homozygous Protein S deficiency Protein C deficiency	Deceased
27	Abdominal	Thrombosis of the portal vein	Acute conditions Maternal SARS-CoV-2 infection Pregnancy	MTHFR A1298C—heterozygous Protein S deficiency Protein C deficiency	Discharged
28	Abdominal	Thrombosis of the inferior vena cava	Acute conditions Major surgery Hemodialysis	MTHFR A1298C—heterozygous MTHFR C677T—heterozygous	Deceased
29	Abdominal	Thrombosis of the inferior vena cava, both renal veins, and the left common iliac vein	Sepsis Maternal Thrombophilia Maternal miscarriage Family history of perinatal stroke	MTHFR A1298C—heterozygous	Deceased
30	Limb	Severe skin ischemia and necrosis	Acute condition Prolonged hospitalization (>3 months)	PAI-1 4G/5G promoter—heterozygous MTHFR A1298C—heterozygous MTHFR C677T—heterozygous Antithrombin deficiency Protein S deficiency Protein C deficiency	Deceased
31	Limb	Severe skin ischemia and necrosis	Acute conditions Major surgeries Sepsis Maternal Thrombophilia	MTHFR A1298C—heterozygous MTHFR C677T—heterozygous PAI-1 4G/5G promoter—heterozygous	Discharged
32	Limb	Thrombosis of the left femoral artery	Acute condition Major surgeries	MTHFR A1298C—heterozygous Antithrombin deficiency Protein S deficiency Protein C deficiency	Deceased
33	Limb	Right upper limb ischemia and necrosis Thrombosis of the inferior vena cava	Maternal SARS-CoV-2 infection Pregnancy	MTHFR A1298C—heterozygous MTHFR C677T—heterozygous Antithrombin deficiency	Discharged
34	Limb	Thrombosis of the right axillary artery	Acute condition Major surgery Catheter placement Maternal thrombophilia	MTHFR C677T—homozygous Antithrombin deficiency Protein S deficiency Protein C deficiency	Discharged
35	Thoracic	Thrombosis of the right pulmonary artery and intracardiac thrombosis	None	MTHFR C677T—homozygous Antithrombin deficiency Protein S deficiency Protein C deficiency	Discharged
36	Thoracic	Intracardiac thrombosis	Acute condition Major surgery Sepsis	PAI-1 4G/5G promoter—homozygous MTHFR A1298C—homozygous Antithrombin deficiency Protein C deficiency	Deceased
37	Thoracic	Intracardiac thrombosis	Acute condition Major surgery Maternal miscarriage	Prothrombin G20210A—heterozygous MTHFR A1298C—heterozygous Antithrombin deficiency Protein S deficiency Protein C deficiency	Deceased
38	Thoracic	Left chylothorax	Acute condition Catheter placement Sepsis	Factor V Leiden—heterozygous MTHFR C677T—heterozygous	Deceased
39	Generalized	Thrombosis of both internal jugular veins, both brachiocephalic veins, the superior vena cava, the inferior vena cava, both common iliac veins, both external iliac veins and both femoral veins	Catheter placement Sepsis Multiple fractures Prolonged CPR Severe chronic illness Prolonged hospitalization (>3 months) Family history of thrombosis	MTHFR C677T—homozygous Antithrombin deficiency	Deceased
40	Generalized	Thrombosis of both internal jugular veins, superior and inferior venae cavae and both femoral veins Bilateral chylothorax	Acute condition Major surgeries Catheter placement Prolonged hospitalization (>3 months)	Prothrombin G20210A—heterozygous MTHFR C677T—heterozygous Protein C deficiency	Discharged

## Data Availability

Data are available on request from the corresponding author.

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
