# Peer review of "Consistent Correlation between MTHFR and Vascular Thrombosis in Neonates—Case Series and Clinical Considerations"

_jcm, 2023, doi:10.3390/jcm12144856_

Round 1
Reviewer 1 Report
The authors can present the topic with a better scenario than the one they presented, so that the idea is clearer, with the use of statistical operations in a professional manner that clarifies the idea for which the research was conducted. An SPSS program was mentioned in the search method, but I did not see where it was used in the results. Mentioning the details of each case can give an idea to the reader, but at the same time, it can create dispersion and loss of the main idea of the research. I hope that the method of drafting the manuscript will be rewritten so that it reflects the efforts of the researchers and gives a clear idea to the reader
Author Response
The authors can present the topic with a better scenario than the one they presented, so that the idea is clearer, with the use of statistical operations in a professional manner that clarifies the idea for which the research was conducted. An SPSS program was mentioned in the search method, but I did not see where it was used in the results. Mentioning the details of each case can give an idea to the reader, but at the same time, it can create dispersion and loss of the main idea of the research. I hope that the method of drafting the manuscript will be rewritten so that it reflects the efforts of the researchers and gives a clear idea to the reader
Our manuscript has been revised accordingly, we tried to reshape it for a better understanding. The statistical part has also been added.
Reviewer 2 Report
This manuscript described in detail on case to case basis effect of MTHFR mutations in Romanian populations. The conclusion by the authors of this case series " MTHFR mutations 40 should be regarded as thrombotic risk factors for our critically ill patients, and screening for MTHFR 41 mutations should be commenced in every admitted patient to intensive care units, thus, prevention 42 of thrombi could be performed" are fully supported by the way research was conducted and presented.
Following are some observations to proceed prior to further publication
1: Introduction need to be more elaborated and focused with addition of more recent research resources
2: Discussion section need to be added before conclusion.
Author Response
Dear Reviewer,
We appreciate that you took the time to review our manuscript, and we are grateful for the advice.
This manuscript described in detail on case to case basis effect of MTHFR mutations in Romanian populations. The conclusion by the authors of this case series " MTHFR mutations 40 should be regarded as thrombotic risk factors for our critically ill patients, and screening for MTHFR 41 mutations should be commenced in every admitted patient to intensive care units, thus, prevention 42 of thrombi could be performed" are fully supported by the way research was conducted and presented.
Following are some observations to proceed prior to further publication
1: Introduction need to be more elaborated and focused with addition of more recent research resources
We have improved our Introduction section.
2: Discussion section need to be added before conclusion.
Regarding the reviewer's comment, our manuscript already has a Discussion section before the Conclusion part.